# PON-Sol2: Prediction of Effects of Variants on Protein Solubility

**DOI:** 10.3390/ijms22158027

**Published:** 2021-07-27

**Authors:** Yang Yang, Lianjie Zeng, Mauno Vihinen

**Affiliations:** 1School of Computer Science and Technology, Soochow University, Suzhou 215006, China; yyang@suda.edu.cn (Y.Y.); 1709404010@stu.suda.edu.cn (L.Z.); 2Collaborative Innovation Center of Novel Software Technology and Industrialization, Nanjing 210000, China; 3Department of Experimental Medical Science, Lund University, BMC B13, SE-221 84 Lund, Sweden

**Keywords:** protein solubility prediction, prediction, machine learning, variation interpretation, artificial intelligence, variation, mutation, PON-Sol2

## Abstract

Genetic variations have a multitude of effects on proteins. A substantial number of variations affect protein–solvent interactions, either aggregation or solubility. Aggregation is often related to structural alterations, whereas solubilizable proteins in the solid phase can be made again soluble by dilution. Solubility is a central protein property and when reduced can lead to diseases. We developed a prediction method, PON-Sol2, to identify amino acid substitutions that increase, decrease, or have no effect on the protein solubility. The method is a machine learning tool utilizing gradient boosting algorithm and was trained on a large dataset of variants with different outcomes after the selection of features among a large number of tested properties. The method is fast and has high performance. The normalized correct prediction rate for three states is 0.656, and the normalized GC2 score is 0.312 in 10-fold cross-validation. The corresponding numbers in the blind test were 0.545 and 0.157. The performance was superior in comparison to previous methods. The PON-Sol2 predictor is freely available. It can be used to predict the solubility effects of variants for any organism, even in large-scale projects.

## 1. Introduction

Genetic variations have numerous effects. The largest portion of known disease-causing and disease-related variations is in protein-coding regions. In variation interpretation, the goal is to detect the harmful variants. There are numerous prediction methods available for this purpose, e.g., [1,2,3,4]. These tools are useful; however, they do not reveal any details about the causative mechanism and thereby of possible countermeasures, such as drugs and others. Other types of tools have been released for predicting alterations to protein properties, such as solubility.

Solubility of a protein is one of its fundamental characteristics [5]. Solubilities vary widely among proteins and protein forms. Proteome-wide analysis of solubility in *Caenorhabditis elegans* indicated that about 75% of proteins appear in cells in abundances close to their solubility limits [6]. There has not been evolutionary pressure to make the proteins more soluble.

Two concepts are related to protein–solvent interactions. Solubility is usually defined as the concentration in which intact protein is in equilibrium with solid phase [7,8,9,10]. Precipitated solubilizable protein in solid phase can be made again soluble by dilution. The other phenomenon is aggregation. When proteins aggregate, they bind together, which is often accompanied by irreversible alteration to conformation, leading to the formation of insoluble high-molecular-weight forms [5].

Protein solubility depends on many factors. Intrinsic properties of the protein, solvent, and additives are important along with physical conditions. Relevant protein factors include the amino acid sequence and its composition, three-dimensional structure, accessibility, and intramolecular interactions within the protein as well as protonation status. Salt bridges, electrostatic and hydrophobic interactions, and weak hydrogen bonds all affect solubility. Whether a protein is monomeric or multimeric has also an effect. Important solvent properties include polarity, its bond and interaction-forming ability, density and included additives and constituents, such as excipients, salts, and organic solvents. Concentrations of these compounds have significant contributions to protein solubility. Of the environmental parameters, pH and temperature can affect both the protein and solvent.

Alterations to proteins can affect their properties. Single amino acid alterations can profoundly alter protein solubility and lead to diseases. Severe complex V deficiency [11] and cataract [12] are examples.

To address the effects of protein variants, some prediction methods have been released. These include CamSol [13], OptSolMut [14], PON-Sol [9], SODA [15], and SolubiS [16] and have been reviewed in [10]. CamSol uses a residue-specific solubility profile. Only the algorithm has been described; no method has been made available. OptSolMut was trained with 137 single and multiple variants affecting solubility or aggregation. Weights were optimized for a scoring function with linear programming. PON-Sol, a random forest-based method, was trained this far on the largest dataset of 406 single amino acid substitutions. It grouped variants into three classes: solubility decreasing and increasing variants and those not affecting solubility. SODA has been recommended to predict variants decreasing solubility [15]. It was developed with PON-Sol data. It can predict in addition to substitutions also effects of insertions and deletions. SoluBis is a tool for the optimization of multiple variants to increase protein solubility [16]. It is based on the detection of aggregation prone segments to modify them. The prediction is a combination of interaction analysis with FoldX [17], aggregation prediction with TANGO [18], and structural analysis with YASARA [19].

We have previously developed several high-performance prediction methods for variation effects, mainly based on machine learning (ML) algorithms. These include pathogenicity/tolerance prediction methods PON-P (Olatubosun et al. 2012) and PON-P2 (Niroula et al. 2015) for filtering harmful variants from sequencing datasets. We developed the first generic variation phenotype severity predictor (Niroula and Vihinen, 2017). To investigate the mechanisms and effects of amino acid substitutions, there are PON-Diso (Ali et al. 2014) for protein disorder, PON-Tstab for protein stability (Yang et al. 2018), PROlocalizer (Laurila and Vihinen, 2011) for protein localization, and PON-Sol (Yang et al. 2016) for protein solubility affecting variants.

Since the publication of PON-Sol, a substantial amount of new cases has been published and warranted the development of an entirely new predictor, PON-Sol2, which has superior performance in comparison to the previous tools. We collected by far the largest set of experimentally verified variants and used them to train an ML predictor and tested it in an independent test dataset. The developed tool can be used for variation interpretation and analysis of the mechanisms in disease-related variants and to design variants for protein engineering, protein crystallization, and other applications.

## 2. Results

We developed a novel machine learning-based predictor for effects of single AASs on protein solubility. We collected a large dataset of over 10,000 cases, which was reduced to 6328 variations to due to class imbalance. There was still imbalance, which was mitigated in the method performance assessment.

The dataset included variants from altogether 77 proteins and represented all substitution types. The distribution of the AASs (Table 1) indicates that leucine (659), alanine (564), and isoleucine (506) are the most common original residues. The most common substitution is by proline (420). The most common amino acid substitutions are L > S (59), L > T (49), L > A (47), and L > E (44) alterations. There were up to 43 variants of a certain substitution type.

### 2.1. Feature Selection and Method Training

We started with 1081 features in the categories of amino acid propensities and characteristics, conservation, variation type, neighborhood features, and chain length. RFE was applied in feature selection. GOSS down samples the instances on the basis of gradients [20]. Instances with small gradients are well trained and have a small training error, while those with large gradients are undertrained. GOSS retains instances with large gradients while performing random sampling on instances with small gradients. EFB reduces the number of features by regrouping mutually exclusive features into bundles and then treating them as a single feature [20]. This is beneficial especially in sparse feature space where many features are (almost) exclusive and very seldom have non-zero values simultaneously. Thus, these kinds of features can be safely bundled.

An initial comparison of random forest, XGBoost, and LightGBM indicated that the gradient boosting algorithms had better performance. LightGBM was chosen due to its speed to train and run; the performance was almost identical with XGBoost. Results for predictors with all features, 100, 50, and 20 features were very similar; however, they were somewhat better when using a smaller number of features.

We tested two architectures for the predictor implementation. In one of them, single predictor distributes the cases to three categories. The other one is combination of two two-layer predictors. The reasoning for testing the two-layer predictor was our earlier experience with variant severity predictor PON-PS [21], and variant stability predictor PON-Stab [22] indicated that a combination of two-layer predictions could be beneficial. For the two-layer three-class classifier we generated binary classifiers for both the layers (Figure 1). For the first layer, we marked the variations increasing or having no effect on solubility as “not-decreasing” and trained a “decreasing/not-decreasing” classifier. The second layer only used the variations increasing and having no effect to train an “increasing/no effect” classifier. We chose to use 20 features per predictor in the two-layer predictor and 30 in the single-layer predictor to have as small a set of features as possible and thereby covering the space of feature distribution better.

When we trained the two-layer three-class LightGBM classifier, RFE was used to select 20 features for each layer. As the predictors shared six features, 34 different features were selected (Table 2 and Table 3). Among them there are 22 amino acid features, 11 neighborhood features, and the length of protein sequence. This method was finally chosen as it showed somewhat better performance than the single three-class predictor, its CPR in 10-fold CV was 0.747 (Table 4 and Table 5).

### 2.2. Performance Assessment

The method performance was assessed according to published guidelines [23,24]; also, the other items of the guidelines were followed. Due to the uneven distribution of cases in the three solubility categories, we normalized the calculated results in Table 4, Table 5 and Table 6. The first figure indicates the number of cases, the second one is for normalized values for cases in categories TP to FN. For performance measures, the first one is without normalization, the latter one is with normalization, and these are the numbers that we compared. Table 4 lists the 10-fold cross-validation (CV) performance for four classifiers. As the scores were equal or better for predictors with smaller feature sets, we chose the 34-feature two-layer three-class classifier and call it PON-Sol2. The two-layer predictor is marginally better than the single-layer one. Its normalized CPR is 0.656, which is significantly improved compared with 0.491 for the original PON-Sol.

**Table 2 ijms-22-08027-t002:** Features selected by RFE for two-layer three-class LightGBM decreasing/not decreasing classifier sorted by importance. Features shared by to the two predictors are underlined.

Rank	Name	Feature	Description
1	FUKS010101	Amino acid feature	Surface composition of amino acids in intracellular proteins of thermophiles (percent) [25]
2	JOND920102	Amino acid feature	Relative mutability [26]
3	PONP800107	Amino acid feature	Accessibility reduction ratio [27]
4	NonPolarAA	Neighborhood feature	Number of nonpolar residues in the neighborhood window
5	PolarAA	Neighborhood feature	Number of polar residues in the neighborhood window
6	QIAN880134	Amino acid feature	Weights for coil at the window position of [28]
7	AA20D.T	Neighborhood feature	Number of threonine residues in the neighborhood window
8	PosAA	Neighborhood feature	Number of positively charged residues in the neighborhood window
9	AA20D.L	Neighborhood feature	Number of leucine residues in the neighborhood window
10	GEOR030102	Amino acid feature	Linker propensity from 1-linker dataset [29]
11	OOBM850102	Amino acid feature	Optimized propensity to form reverse turn [30]
12	length	Protein type feature	Number of amino acids in the protein sequence
13	AA20D.I	Neighborhood feature	Number of isoleucine residues in the neighborhood window
14	AA20D.P	Neighborhood feature	Number of proline residues in the neighborhood window
15	KOSJ950115	Amino acid feature	Context-dependent optimal substitution matrices for all residues [31]
16	ARGP820102	Amino acid feature	Signal sequence helical potential [32]
17	PRAM820103	Amino acid feature	Correlation coefficient in regression analysis [33]
18	AA20D.V	Neighborhood feature	Number of valine residues in the neighborhood window
19	ZIMJ680104	Amino acid feature	Isoelectric point [34]
20	CHOP780209	Amino acid feature	Normalized frequency of C-terminal beta-sheet [35]

**Table 3 ijms-22-08027-t003:** Features selected by RFE for two-layer three-class LightGBM increasing/no effect classifier sorted by importance. Features shared by to the two predictors are underlined.

Rank	Name	Feature	Description
1	VASM830102	Amino acid feature	Relative population of conformational state C [36]
2	PRAM820103	Amino acid feature	Correlation coefficient in regression analysis [33]
3	DAYM780201	Amino acid feature	Relative mutability [37]
4	ChargedAA	Neighborhood feature	Number of charged residues in the neighborhood window
5	DOSZ010102	Amino acid feature	Normalized version of SM_SAUSAGE [38]
6	NonPolarAA	Neighborhood feature	Number of nonpolar residues in the neighborhood window
7	PRAM820101	Amino acid feature	Intercept in regression analysis [33]
8	BROC820102	Amino acid feature	Retention coefficient in HFBA [39]
9	PolarAA	Neighborhood feature	Number of polar residues in the neighborhood window
10	MIRL960101	Amino acid feature	Statistical potential derived by the maximization of the harmonic mean of Z scores [40]
11	AA20D.D	Neighborhood feature	Number of aspartic acid residues in the neighborhood window
12	VASM830101	Amino acid feature	Relative population of conformational state A [36]
13	SUYM030101	Amino acid feature	Linker propensity index [41]
14	length	Protein type feature	Number of amino acids in the protein sequence
15	FASG760103	Amino acid feature	Optical rotation [42]
16	CHOP780213	Amino acid feature	Frequency of the 2nd residue in turn [35]
17	AA20D.L	Neighborhood feature	Number of leucine residues in the neighborhood window
18	LIFS790102	Amino acid feature	Conformational preference for parallel beta-strands [43]
19	PosAA	Neighborhood feature	Number of positively charged residues in the neighborhood window
20	AA20D.G	Neighborhood feature	Number of glycine residues in the neighborhood window

The performance figures are shown separately for the three categories, and there are clear differences between them. Normalized positive predictive value is the best for solubility decreasing cases (0.781) followed by solubility increasing variants (0.714), while those having no effect have the lowest score (0.534). In the case of normalized NPV, the three categories are predicted almost equally well (0.855 to 0.891). Sensitivity again shows big differences; this time, the solubility decreasing cases have the lowest score. Specificity values, although variable, are closer to each other than those for sensitivity. Normalized CPR of 0.656 shows good performance. Note that a random three-class predictor would have a score of 0.333. The normalized GC^2^ score is 0.312.

**Table 4 ijms-22-08027-t004:** Comparison of different three-class LightGBM classifier designs on 10-fold cross-validation.

Performance Measure		Predictor			
		Single Three-Class Classifier		Two-Layer Three-Class Classifier	
		All Features	30 Features Selected by RFE	All Features	34 Features Selected by RFE
TP	−	257.1/177.2	253.6/174.8	249.6/172.1	249.2/171.8
	no	135.3/135.3	138.4/138.4	139.7/139.7	142.4/142.4
	+	30.9/63.5	30.6/62.9	31.6/64.9	31.9/65.5
TN	−	238.4/323.8	236.6/320.7	250.9/340.5	249.0/337.3
	no	303.3/268.9	301.5/268.9	293.2/257.4	296.1/261.5
	+	448.2/362.0	451.1/365.2	443.4/357.5	445.0/359.6
FP	−	48.4/62.0	50.2/65.1	35.9/45.3	37.8/48.5
	no	70.4/116.9	72.2/116.9	80.5/128.4	77.6/124.3
	+	24.5/23.8	21.6/20.6	29.3/28.3	27.7/26.2
FN	−	22.7/15.7	26.2/18.1	30.2/20.8	30.6/21.1
	no	57.6/57.6	54.5/54.5	53.2/53.2	50.5/50.5
	+	63.0/129.4	63.3/130.0	62.3/128.0	62.0/127.4
PPV	−	0.842/0.742	0.835/0.729	0.875/0.793	0.869/0.781
	no	0.657/0.536	0.658/0.543	0.635/0.521	0.647/0.534
	+	0.563/0.730	0.586/0.752	0.520/0.696	0.538/0.714
NPV	−	0.913/0.954	0.901/0.947	0.893/0.942	0.891/0.941
	no	0.841/0.824	0.847/0.832	0.847/0.829	0.855/0.838
	+	0.877/0.737	0.877/0.738	0.877/0.736	0.878/0.739
Sensitivity	−	0.919/0.919	0.906/0.906	0.892/0.892	0.891/0.891
	no	0.701/0.701	0.717/0.717	0.724/0.724	0.738/0.738
	+	0.329/0.329	0.326/0.326	0.336/0.336	0.340/0.340
Specificity	−	0.831/0.839	0.825/0.831	0.875/0.883	0.868/0.874
	no	0.812/0.697	0.807/0.697	0.785/0.667	0.792/0.678
	+	0.948/0.938	0.954/0.947	0.938/0.927	0.941/0.932
CPR		0.747/0.650	0.746/0.650	0.743/0.651	0.747/0.656
GC^2^		0.317/0.298	0.309/0.289	0.322/0.313	0.323/0.312

### 2.3. Performance on Blind Test Set

The obtained cases were initially partitioned to generate a blind test set. This dataset was tested only after the training phase was finished. In the generation of these data, we took into account that in the levoglucosan kinase and β-lactamase, there were several variants that changed the same original residue. To avoid bias in testing, data partition was made so that all substitutions within a position were either in the training or test set.

The blind test set contained 662 variants, of which 338 decreased solubility, 237 increased, and 87 had no effect on solubility. The results in Table 5 are well in line with those for CV in Table 4. The overall scores are somewhat smaller, but otherwise, the results are very similar to CV results. Normalized CPR was 0.545 and normalized CG^2^ 0.157. The differences in individual measures for the three solubility categories are very similar to CV data, indicating that e.g., the PPV and sensitivity of solubility increasing cases are more difficult to predict than the two other classes.

**Table 5 ijms-22-08027-t005:** Comparison of different three-class LightGBM classifier designs on blind test dataset.

Performance Measure		Predictor			
		Direct Three-Class Classifier		Two-Layer Three-Class Classifier	
		All Features	30 Features Selected by RFE	All Features	34 Features Selected by RFE
TP	−	288.0/201.9	282.0/197.7	272.0/190.7	271.0/190.0
	no	154.0/154.0	151.0/151.0	160.0/160.0	159.0/159.0
	+	11.0/30.0	7.0/19.1	10.0/27.2	14.0/38.1
TN	−	235.0/341.9	238.0/343.2	247.0/355.6	258.0/368.3
	no	329.0/303.5	323.0/293.2	313.0/288.3	319.0/298.5
	+	551.0/451.5	541.0/442.4	544.0/445.1	529.0/431.3
FP	−	89.0/132.1	86.0/130.8	77.0/118.4	66.0/105.7
	no	96.0/170.5	102.0/180.8	112.0/185.7	106.0/175.5
	+	24.0/22.5	34.0/31.6	31.0/28.9	46.0/42.7
FN	−	50.0/35.1	56.0/39.3	66.0/46.3	67.0/47.0
	no	83.0/83.0	86.0/86.0	77.0/77.0	78.0/78.0
	+	76.0/207.0	80.0/217.9	77.0/209.8	73.0/198.9
PPV	−	0.764/0.605	0.766/0.602	0.779/0.617	0.804/0.643
	no	0.616/0.475	0.597/0.455	0.588/0.463	0.600/0.475
	+	0.314/0.571	0.171/0.376	0.244/0.485	0.233/0.472
NPV	−	0.825/0.907	0.810/0.897	0.789/0.885	0.794/0.887
	no	0.799/0.785	0.790/0.773	0.803/0.789	0.804/0.793
	+	0.879/0.686	0.871/0.670	0.876/0.680	0.879/0.684
Sensitivity	−	0.852/0.852	0.834/0.834	0.805/0.805	0.802/0.802
	no	0.650/0.650	0.637/0.637	0.675/0.675	0.671/0.671
	+	0.126/0.126	0.080/0.080	0.115/0.115	0.161/0.161
Specificity	−	0.725/0.721	0.735/0.724	0.762/0.750	0.796/0.777
	no	0.774/0.640	0.760/0.619	0.736/0.608	0.751/0.630
	+	0.958/0.953	0.941/0.933	0.946/0.939	0.920/0.910
CPR		0.684/0.543	0.665/0.517	0.668/0.532	0.671/0.545
GC^2^		0.173/0.150	0.165/0.141	0.162/0.141	0.181/0.157

### 2.4. Comparison to Other Tools

Of the previous methods, only SODA and the original PON-Sol could be compared with PON-Sol2. SODA is designed to predict in addition to AASs also insertions and deletions. It is a binary classifier that predicts variations increasing solubility or decreasing solubility. The score of SODA is calculated from the weighted sum of five score differences.

In an effort to test whether SODA could be used in three-state prediction, we applied different thresholds at 5, 10, and 17 to include a class for variants with no effect on solubility. The threshold at 17 gave the best result (Table 6); however, the normalized CPR is only 0.356, and the normalized GC^2^ is 0.016. The corresponding scores are 0.389 and 0.011 for PON-Sol. PON-Sol2 is significantly better, the scores being 0.545 and 0.157, respectively. Furthermore, PON-Sol2 is clearly a more balanced predictor, the two other tools showing larger differences for the scores of the different types of solubility effects.

**Table 6 ijms-22-08027-t006:** Comparison of the prediction performance of PON-Sol2 with SODA and PON-Sol2.

Performance Measure							
		PON-Sol	SODA	SODA (5 as Threshold)	SODA (10 as Threshold)	SODA (17 as Threshold)	PON-Sol2
TP	−	89.0/62.4	165.0/115.7	66.0bb/46.3	33.0/23.1	23.0/16.1	271.0/190.0
	no	108.0/108.0	0.0/0.0	180.0/180.0	210.0/210.0	226.0/226.0	159.0/159.0
	+	39.0/106.2	22.0/59.9	6.0/16.3	5.0/13.6	4.0/10.9	14.0/38.1
TN	−	263.0/390.6	103.0/140.9	281.0/412.0	308.0/447.7	316.0/462.6	258.0/368.3
	no	280.0/301.5	425.0/474.0	181.0/161.3	106.0/96.6	61.0/54.9	319.0/298.5
	+	355.0/295.5	321.0/271.7	452.0/380.3	496.0/413.5	538.0/446.6	529.0/431.3
FP	−	61.0/83.4	221.0/333.1	43.0/62.0	16.0/26.3	8.0/11.4	66.0/105.7
	no	145.0/172.5	0.0/0.0	244.0/312.7	319.0/377.4	364.0/419.1	106.0/175.5
	+	220.0/178.5	254.0/202.3	123.0/93.7	79.0/60.5	37.0/27.4	46.0/42.7
FN	−	249.0/174.6	173.0/121.3	272.0/190.7	305.0/213.9	315.0/220.9	67.0/47.0
	no	129.0/129.0	237.0/237.0	57.0/57.0	27.0/27.0	11.0/11.0	78.0/78.0
	+	48.0/130.8	65.0/177.1	81.0/220.7	82.0/223.4	83.0/226.1	73.0/198.9
PPV	−	0.593/0.428	0.427/0.258	0.606/0.428	0.673/0.468	0.742/0.585	0.804/0.643
	no	0.427/0.385	nan/nan	0.425/0.365	0.397/0.357	0.383/0.350	0.600/0.475
	+	0.151/0.373	0.080/0.229	0.047/0.149	0.060/0.184	0.098/0.284	0.233/0.472
NPV	−	0.514/0.691	0.373/0.537	0.508/0.684	0.502/0.677	0.501/0.677	0.794/0.887
	no	0.685/0.700	0.642/0.667	0.761/0.739	0.797/0.782	0.847/0.833	0.804/0.793
	+	0.881/0.693	0.832/0.605	0.848/0.633	0.858/0.649	0.866/0.664	0.879/0.684
Sensitivity	−	0.263/0.263	0.488/0.488	0.195/0.195	0.098/0.098	0.068/0.068	0.802/0.802
	no	0.456/0.456	0.000/0.000	0.759/0.759	0.886/0.886	0.954/0.954	0.671/0.671
	+	0.448/0.448	0.253/0.253	0.069/0.069	0.057/0.057	0.046/0.046	0.161/0.161
Specificity	−	0.812/0.824	0.318/0.297	0.867/0.869	0.951/0.944	0.975/0.976	0.796/0.777
	no	0.659/0.636	1.000/1.000	0.426/0.340	0.249/0.204	0.144/0.116	0.751/0.630
	+	0.617/0.623	0.558/0.573	0.786/0.802	0.863/0.872	0.936/0.942	0.920/0.910
CPR		0.356/0.389	0.282/0.247	0.381/0.341	0.375/0.347	0.382/0.356	0.671/0.545
GCC		0.010/0.011	nan/nan	0.041/0.045	0.022/0.022	0.016/0.016	0.181/0.157

### 2.5. Large-Scale Variant Prediction

As PON-Sol2 is a fast predictor, it allows large-scale analyses of solubility effects, such as protein-wide effects. Figure 2 shows predictions for all possible single amino acid substitutions in the Bruton tyrosine kinase (BTK) kinase domain [44]. Loss of function variations in BTK cause X-linked agammaglobulinemia (XLA), which is a primary immunodeficiency due to a block in the B cell maturation pathway. BTK is a central signaling molecule during B cell development, and its activity is crucial for maturation of the cells. Gain of function variants in the B cell receptor signaling pathway, where BTK is involved, appear in B cell malignancies, such as chronic lymphocytic leukaemia and Waldenström macroglobulinemia. XLA-causing variants are collected to BTKbase [45]; there are currently over 1800 variants, amino acid substitutions being among the most common alterations. However, the effects of the variants on solubility are not known, apart from some individual cases.

Figure 2A–C shows predictions for alterations that increase, decrease, or have no effect on solubility. In addition, the tolerance/pathogenicity predictions were obtained for all substitutions with a reliable PON-P2 predictor [4]. The method classifies variants in three categories: pathogenic, benign, and unclassified variants. Colour coding was used to indicate the numbers for predicted disease-causing variants, which are shown in Figure 2D. The numbers of predicted solubility decreasing alterations in Figure 2C imply certain correlation with 2D. Since many effects lead to XLA, we cannot even expect to see a 1:1 correlation with one effect. Solubility is just one of the effects of variations that lead to XLA. There are substantially more solubility-affecting variants in many positions where there are many disease-related variants.

### 2.6. PON-Sol2 Web Application

PON-Sol2 web application is freely available at http://structure.bmc.lu.se/PON-Sol2/ (accessed on 26 July 2021) and http://139.196.42.166:8010/ (accessed on 26 July 2021). There is a user-friendly web interface. It accepts variations in two formats: sequence and identifier formats. Sequence submission is for FASTA format amino acid sequence(s) and amino acid substitutions in it (them). Identifier submission requires amino acid substitutions and one of UniProtKB/Swiss-Prot accession ID, Entrez gene ID, or Ensemble ID. For these submissions, PON-Sol2 makes predictions only for variations leading to amino acid substitutions. Batch submission including all variants and proteins of interest is accepted and recommended. PON-Sol2 provides a complete report, which is sent to the user by email when ready.

## 3. Discussion

Amino acid substitutions can have widely differing effects; for a recent discussion of protein function affecting effects and mechanisms, see [46]. Solubility is one factor that can contribute to changes to function. Some single amino acid changes are responsible for substantially decreased or improved solubility. Many proteins are expressed in concentrations close to their maximal solubility [6]. Predictions of solubility effects have several applications. They can be used in variation interpretation in diseases. Protein engineering could benefit from reliable predictions of solubility alterations due to variations. Knowing which residue to change and how the properties could be changed can be used to design proteins that could be expressed in large quantities in various host organisms and expression systems.

Protein crystallization is another application area for more soluble proteins. X-ray crystallography is based on highly ordered crystals. Despite extensive trials, all proteins are not amenable for crystallization. There are many reasons; one of the common ones is that the protein is not soluble in the required concentrations. This could be improved by modifying the protein to increase its solubility. Even Nuclear Magnetic Resonance (NMR) studies of protein structures in solution require high protein concentrations and would thus benefit from solubility-increasing variants.

PON-Sol2 shows clearly improved performance in comparison to the original PON-Sol. This is expected from much larger training data, 5666 vs. 406 cases. Despite the substantial growth of data, it would still be possible to increase the performance with even bigger sets of experimental cases originating from larger number of proteins. We would need data for variations in different types of proteins and in different structural and sequence contexts.

The single and two-layer implementations did not show marked differences in prediction performance. When training the method, the selected features had relatively small significance scores, unlike in the pathogenicity/tolerance predictor PON-P2 [4]. It was possible to reduce the number of features to 34 in the two-layer predictor. The performance improvement in comparison to a predictor with all the features was not very high. The major benefit comes from the fact that with the limited set of features, representativeness of the variant space may be significantly better. The method is fast and reliable and facilitates predictions even for large numbers of variants.

## 4. Materials and Methods

### 4.1. Data

The dataset contains all the original PON-Sol cases of 443 single amino acid substitutions in 71 proteins [9]. In addition, we collected based on an extensive literature search 10,758 variants in six additional proteins: 10 amino acid substitutions (AASs) in ThreeFoil [47], 76 in *Escherichia coli* cytotoxin [48], 6 in aminoacyl-tRNA synthetase [49], 3 in α-spectrin SH3 domain [50], 6298 in levoglucosan kinase [51], and 4365 inTEM-1 β-lactamase [51]. Altogether, there were 11,201 AASs in 77 proteins. We paid special attention to detect cases affecting (or not) solubility. The literature for aggregation-related variants is substantially larger.

The variants were grouped into three categories: solubility increasing and decreasing cases and those having no effect on solubility. The classifications were obtained from original publications, except for the last two proteins. For those, solubility scores of yeast surface display (YSD) and twin-arginine translocation (Tat) were considered. In the end, only YSD data were used, since the Tat data contained lots of false negatives. As the threshold, we used 0.15 in YSD data to define the three types of variations [51]. Since the dataset was heavily biased toward solubility-decreasing cases, we randomly excluded solubility decreasing cases in levoglucosan kinase and TEM-1 β-lactamase data, so that we finally had 6328 variations, 3136 of which decreased solubility, 1026 increased solubility, and 2166 had no effect, with the ratio of 1:0.69:0.34.

The variants were randomly partitioned into training and test sets. In the case of levoglucosan kinase and TEM-1 β-lactamase, the division was made position wise; i.e., all variations in a certain position were used either for training or testing. In total, 5666 variants (2798 solubility decreasing, 1929 increasing, and 939 without effect on solubility) were used for training. The blind test contained 662 variants, of which 338 decreased solubility, 237 increased solubility, and 87 had no effect.

The datasets are freely available in VariBench database [52,53] at http://structure.bmc.lu.se/VariBench/ponsol2.php (accessed on 26 July 2021).

### 4.2. Features

We collected as large a set of features as possible, since it is not possible to know beforehand which features and their combinations are useful for predictions. We started with 1081 features of which 617 were amino acid features, 2 were conservation features, 436 were variation type features, 25 were neighborhood features, and 1 was a protein-type feature.

Amino acid features were from the AAindex database (accessed on 2 March 2020) [54] and selected as previously described for PON-P2 [4] and PON-MMR2 [55] predictors. Conservation features included the SIFT score and the number of hits. Protein sequences were used as queries in a DIAMOND v0.9.29 [56] search against the NCBI (accessed on 2 March 2020) bnon-redundant database to find homologous sequences. Then, the sequences with percentage of identical matches greater than 90% were aligned by BLAST [57] and used to calculate the SIFT score (v6.2.1) [58] for each variation.

Variation-type features contained a 20 × 20 matrix for substitutions. Another 6 × 6 matrix was built according to amino acid grouping to hydrophobic (V, I, L, F, M, W, Y, and C), negatively charged (D, E), positively charged (R, K, H), conformational (G, P), polar (N, Q, S), and others (A, T), as previously described [59].

Neighborhood features were defined with a 20-dimensional vector of neighboring residues that counts the occurrences of each amino acid type within a window of 23 positions; the variant position was in the middle. Features for NonPolarAA, PolarAA, ChargedAA, PosAA, and NegAA denote the numbers of nonpolar, polar, charged, positively charged, and negatively charged neighborhood residues [60], respectively. The protein-type feature is for the length of the protein sequence.

### 4.3. Algorithms

Three machine learning algorithms were initially tested—random forests [61], XGBoost [62], and LigthGBM [20]. All the algorithms were implemented in Python in the standard scikit-learn package [63].

Random forests is an ensemble algorithm. It applies several decision trees on the subset of the dataset and uses the average accuracy of each decision tree to improve the performance and to reduce overfitting. The gradient boosting model evaluates the output features based on the combination output result of weak prediction learner models. It minimizes a loss function to optimize the model. Sequential models are constructed using the decision trees until maximum accuracy is achieved.

XGBoost and LigthGBM are implementations of gradient boosting and based on decision trees. Initial results for LightGBM and XGBoost were similar and better than for random forests. As a result of similar performance, we chose LightGBM, which is faster due to utilizing Gradient-based One-Side Sampling (GOSS) and Exclusive Feature Bundling (EFB).

### 4.4. Feature Selection

Features were chosen with Recursive Feature Elimination (RFE) [64]. In the beginning of feature selection, RFE was used to train a classifier with all features and to define the importance of all features. Then, the least important feature was eliminated. This was repeated recursively to reduce the features until the specified number was reached. The numbers of features tested were all features, 100, 50, and 20 features. Then, predictors were trained with the selected features and tested. As the results were very similar for different numbers of features, we chose the predictor with the smallest number of features to avoid the curse of dimensionality.

### 4.5. Performance Evaluation

For single group classification of solubility, measures were determined as previously suggested [23,24]. We included positive predictive value (*PPV*), negative predictive value (*NPV*), sensitivity, and specificity. Of the recommended measures, accuracy and Matthews correlation coefficient were not used, as the tool predicts three classes. *TP, TN, FP*, and FN represent the numbers of true positives, true negatives, false positives, and false negatives, respectively. The standard performance measures were computed by using the following Equations (1)–(7):(1)$$PPV=\frac{TP}{TP+FP}$$
(2)$$NPV=\frac{TM}{TN+FN}$$
(3)$$Sensitivity=\frac{TP}{TP+FP}$$
(4)$$Specificity=\frac{TN}{TN+FP}\;.$$

To evaluate the overall performance, the correct prediction ratio (*CPR*) and the generalized squared correlation (*GC^2^*) were used, the latter has been suggested for K-class classification [65]. *CPR* is the percentage of correct predictions. *GC^2^* represents the correlation coefficient of the classification ranging from 0 to 1; larger values show better performance. *CPR* and *GC^2^* are defined as
(5)$$CPR=\frac{\sum_{i}z_{ii}}{N},\;\mathrm{and}$$
(6)$$GC^{2}=\frac{\sum_{ij}\frac{{\left(z_{ij}-e_{ij}\right)}^{2}}{e_{ij}}}{N\left(K-1\right)},$$
where *K* is the number of classes and *N* is the number of cases. *z_ij_* represents the number of cases of class *i* to class *j, x_i_ = ∑_j_z_ij_* represents the number of the inputs associated with class *I*, and *y_i_ = ∑_j_z_ij_* represents the number of inputs predicted to be in class *i*. The expected number of cases in cell *i, j* of the confusion matrix can be defined as
(7)$$eij=\frac{x_{i}\times y_{j}}{N}.$$

As the numbers of variants were not balanced in the three solubility categories, the values were normalized to allow the calculation of reliable performance measures.

## Figures and Tables

**Figure 1 ijms-22-08027-f001:**
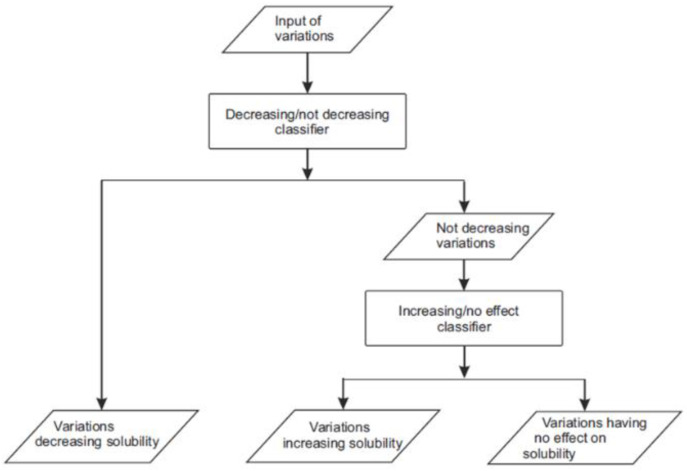
Scheme for two-layer three-class classifier.

**Figure 2 ijms-22-08027-f002:**
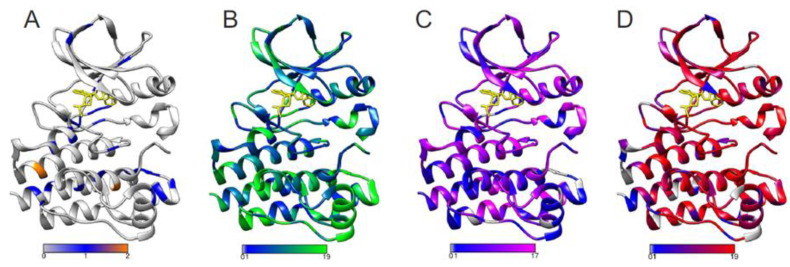
Predicted solubility and disease-related variations in BTK kinase domain (PDB id 5p9j (69), covalent inhibitor at the ATP binding site is in yellow. (**A**) Numbers of variations increasing solubility, (**B**) numbers of variations having no effect on solubility, (**C**) numbers of variants decreasing solubility, and (**D**) numbers of XLA-causing variants. Predictions were made for all 19 single amino acid substitutions at every position. bPathogenicity-related variants were predicted with PON-P2. Keys in the bottom show the numbers of variants predicted to have the effect.

**Table 1 ijms-22-08027-t001:** Distribution of amino acid residues in the dataset. The original amino acids are in rows and variant residues are in columns.

	A	C	D	E	F	G	H	I	K	L	M	N	P	Q	R	S	T	V	W	Y	Total
**A**	0	20	37	30	27	32	29	35	25	34	11	30	43	35	40	29	28	35	17	27	564
**C**	16	0	2	5	1	1	5	5	6	5	4	4	6	3	2	11	2	1	1	3	83
**D**	23	22	0	28	26	19	16	25	23	30	10	29	33	20	33	29	27	14	18	13	438
**E**	29	21	29	0	18	25	10	20	22	25	13	18	24	18	19	23	21	19	22	17	393
**F**	30	11	16	9	0	17	15	6	10	6	2	13	20	14	18	7	18	6	11	8	237
**G**	29	16	25	23	22	0	27	27	16	33	9	18	38	17	31	23	30	12	20	22	438
**H**	13	5	7	3	8	8	0	11	6	6	5	7	10	10	6	11	11	10	6	10	153
**I**	38	19	35	29	11	33	31	0	27	17	15	20	39	30	36	36	22	26	19	23	506
**K**	26	8	18	29	12	18	18	10	0	15	9	14	24	18	23	18	14	17	6	8	305
**L**	47	33	35	44	21	41	36	26	41	0	30	37	25	23	34	59	49	28	21	29	659
**M**	15	13	13	11	9	12	14	14	8	21	0	16	15	17	10	12	5	10	9	9	233
**N**	10	11	13	14	9	14	9	7	9	10	6	0	16	8	9	7	10	10	7	6	185
**P**	14	11	12	14	7	17	21	10	7	11	7	6	0	8	11	14	9	21	8	8	216
**Q**	14	12	14	11	9	12	12	8	14	12	4	11	15	0	14	18	15	14	5	11	225
**R**	26	21	25	21	22	22	27	18	17	23	11	21	26	29	0	28	28	28	17	14	424
**S**	18	8	13	9	11	15	7	6	6	9	4	11	8	9	16	0	15	10	9	11	195
**T**	26	13	17	20	15	25	17	12	12	22	21	24	25	17	22	32	0	20	11	13	364
**V**	17	19	21	18	13	19	25	25	24	23	15	22	32	19	33	42	26	0	14	15	422
**W**	9	5	3	9	6	4	1	2	7	4	3	2	6	4	5	6	5	5	0	2	88
**Y**	14	7	11	13	13	15	11	9	11	14	6	6	15	7	15	9	10	12	2	0	200
**total**	414	275	346	340	260	349	331	276	291	320	185	309	420	306	377	414	345	298	223	249	6328

## Data Availability

PON-Sol2 predictor is available at available at http://structure.bmc.lu.se/PON-Sol2/ (accessed on 26 July 2021) and http://139.196.42.166:8010/ (accessed on 26 July 2021). The program is available at https://github.com/XDcat/PON-Sol2 (accessed on 26 July 2021). Data used for training and testing the method are at VariBench at http://structure.bmc.lu.se/VariBench/ponsol2.php (accessed on 26 July 2021).

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
