# Peer review of "PON-Sol2: Prediction of Effects of Variants on Protein Solubility"

_ijms, 2021, doi:10.3390/ijms22158027_

Round 1
Reviewer 1 Report
Solubility is a central protein property and when reduced can lead to diseases. The authors have developed a prediction method, PON-Sol2, to identify amino acid substitutions that increase, decrease of have no effect on the protein solubility. They employed machine learning model utilizing gradient boosting algorithm and was trained on a large data set of variants with different outcomes after selection of features among large number of tested properties. The model showed very high performance in comparison with existing models, probably due to use of a large number of experimental data.
In terms of very high prediction performance, this paper may be worth being published in IJMS, but I have some major concerns.
1) I could not access the prediction tool at http://structure.bmc.lu.se/PON-Sol2/ and at http://139.196.42.166:8010/.
2) In Table 6, there is no PonSol data. PonSol2 are duplicated.
3)Figure 2 is very hard to understand. In particular, numbers and color gradients are not understandable. I hope the authors have more intelligible explanation.
4)The authors started with 1081 features of which 617 were amino acid features, two conservation features, 436 variation type features, 25 neighbourhood features and one protein type feature. Readers would like to know how much 436 variation type features and 25 neighbourhood feature contribute to prediction performance, respectively. Can the authors explain the contribution of the above two features to prediction?
5) The authors' model obtained remarkably high prediction performance compared to existing models. Readers would like to know how the size of experimental data increases the prediction performance. The authors should show a figure that explains the relationship between the experimental data size and prediction performance.
Author Response
1. PON-Sol2 performed better than PON-Sol and SODA. Is this simply because PON-Sol2 learned larger data? If so, it is trivial. The authors should elucidate and describe the reason why PON-Sol2 was better.
RESPONSE: Many factors contribute to the increased performance of PON-Sol2: larger data, better learning algorithm, using RFE for selecting features, and 2-layer classification. These are discussed in the text. The attached file indicates that amount of data is not directly correlated to the predictor performance.
2. Amino acid mutations are generally harmful. Therefore, it is usual that there are much of solubility decreasing data. I wonder why the authors did not correct this imbalance. For the solubility decreasing data, the performance is the best (Table 4). Is this simply because of the data amount?
RESPONSE: The ratio of harmful variants varies depending on the protein, see e.g. PMID: 25777788 and PMID: 29024182 as well as references therein. There are many proteins in which most variants are tolerated and without phenotype. In regards to the imbalance in trained classed, we wanted to include as many cases into the training set as possible since the space of variants and environments is large. We normalized the performance scores to mitigate differences in the sizes of sets.
Solubility increasing cases are rather rare, thus there are not that many known examples. Differences in the performances of solubility increasing and decreasing predictions originate from different mechanisms, sequence and structure contexts etc., therefore we do not expect them to show exactly similar prediction performance.
3. The performance of cross variation (Table 4) is significantly better than the results of blind test (Table 5). Because the cross variation is an artificial blind test, this big difference needs some explanations. The samples of blind test were very different from the learning data set?
RESPONSE: As described in 4.1, for the generation of test set, the data was divided according to position, which makes the proportional distribution of the different categories slightly different, resulting in a difference in performance. It is normal that cross validation results are better than those for blind test data. This phenomenon has seen before with other variation interpretation predictors as well as with predictors trained for other purposes. This is exactly why we have included also blind test set.
4. I don’t understand the correlation of Fig.2C and Fig. 2D. The authors are required a better presentation. It is truth that the effect of substitution is caused by the original and new amino acids, and the mutational site. Only presenting the mutational site looks odd.
RSPONSE: Figure 2 visualizes the numbers of different types of solubility affecting variants positionwise. We are not aware of any other visualization that would convey the same information (the figure summarizes thousands of variants). This kind of summaries have been published already before in some other articles where large scale predictions are visualized.
5. Simply speaking, substitutions of hydrophilic amino acid to hydrophobic one may decrease the solubility. However, as mentioned at the point 4, the mutational site affects the result. I suppose how the method learns this locational effect is a good representation to show its capability. For example, how many (hydrophobic->hydrophilic) substitutions gain the solubility and for such cases how many the method predicted correctly? How was the inverse case? (hydrophilic->hydrophobic substitutions may lose the solubility)
RESPONSE: The tested features contained more than 100 hydropathy scales. None of them was among the selected features incidating that these parameters cannot explain the effects on protein. This is understandable as an average human protein is some 350 amino acids long and change in just one group within the protein does not have major overall impact.

Reviewer 2 Report
The authors developed a method to predict protein solubility change due to an amino acid substitution (PON-Sol2). They insisted the method was better than the other methods they previously developed. I suppose protein solubility is a crucial issue in protein science, but I would like to pointed out a couple of concerns on the paper.
- PON-Sol2 performed better than PON-Sol and SODA. Is this simply because PON-Sol2 learned larger data? If so, it is trivial. The authors should elucidate and describe the reason why PON-Sol2 was better.
- Amino acid mutations are generally harmful. Therefore, it is usual that there are much of solubility decreasing data. I wonder why the authors did not correct this imbalance. For the solubility decreasing data, the performance is the best (Table 4). Is this simply because of the data amount?
- The performance of cross variation (Table 4) is significantly better than the results of blind test (Table 5). Because the cross variation is an artificial blind test, this big difference needs some explanations. The samples of blind test were very different from the learning data set?
- I don’t understand the correlation of Fig.2C and Fig. 2D. The authors are required a better presentation. It is truth that the effect of substitution is caused by the original and new amino acids, and the mutational site. Only presenting the mutational site looks odd.
- Simply speaking, substitutions of hydrophilic amino acid to hydrophobic one may decrease the solubility. However, as mentioned at the point 4, the mutational site affects the result. I suppose how the method learns this locational effect is a good representation to show its capability. For example, how many (hydrophobic->hydrophilic) substitutions gain the solubility and for such cases how many the method predicted correctly? How was the inverse case? (hydrophilic->hydrophobic substitutions may lose the solubility)
Author Response
1. I could not access the prediction tool at http://structure.bmc.lu.se/PON-Sol2/ and at http://139.196.42.166:8010/.
RESPONSE: The services are online, but there may have been some downtime. The server in Lund is under modifications, the given address will be permanent, in the meantime it can be used from https://structure-next.med.lu.se/PON-Sol2/ There is a link from the final address to the termporary one. The second website can be accessed in China, there may have been temporary problems with access (outside of our control).
2. In Table 6, there is no PonSol data. PonSol2 are duplicated.
RESPONSE:There is an error in the title of the table: the first column should be “PON-Sol”. Thank you for careful reading.
3. Figure 2 is very hard to understand. In particular, numbers and color gradients are not understandable. I hope the authors have more intelligible explanation.
RESPONSE: See our reply to Reviewer 1.
4. The authors started with 1081 features of which 617 were amino acid features, two conservation features, 436 variation type features, 25 neighbourhood features and one protein type feature. Readers would like to know how much 436 variation type features and 25 neighbourhood feature contribute to prediction performance, respectively. Can the authors explain the contribution of the above two features to prediction?
RESPONSE: We cannot address this question as each feature was evaluated separately, not as a group of features. Some neighborhood and variation type features are among the selected ones.
5. The authors' model obtained remarkably high prediction performance compared to existing models. Readers would like to know how the size of experimental data increases the prediction performance. The authors should show a figure that explains the relationship between the experimental data size and prediction performance.
RESPONSE: See the attached table. The relationship between amount of training data and method performance is not linear.
Round 2
Reviewer 1 Report
It is improved.